# PECAM-1/Thrombus Ratio Correlates with Blood Loss during Off-Pump Coronary Artery Bypass Grafting (OPCAB) Surgery: A Preliminary Study

**DOI:** 10.3390/ijms241713254

**Published:** 2023-08-26

**Authors:** Natalia Bielicka, Adrian Stankiewicz, Tomasz Misztal, Szymon Kocańda, Ewa Chabielska, Anna Gromotowicz-Popławska

**Affiliations:** 1Department of Biopharmacy and Radiopharmacy, Medical University of Bialystok, 15-222 Bialystok, Poland; ewa.chabielska@umb.edu.pl (E.C.); anna.gromotowicz@umb.edu.pl (A.G.-P.); 2Department of Cardiosurgery, Medical University of Bialystok, 15-276 Bialystok, Poland; stankiewiczad@wp.pl (A.S.); kardiochirurgia@umb.edu.pl (S.K.); 3Department of Physical Chemistry, Medical University of Bialystok, 15-089 Bialystok, Poland; tomasz.misztal@umb.edu.pl

**Keywords:** platelet, PECAM-1, bleeding, OPCAB, thrombus, fibrin, fibrinolysis

## Abstract

Platelet endothelial cell adhesion molecule 1 (PECAM-1) is considered an antiplatelet molecule. Previously, we introduced a new parameter called the PECAM-1/thrombus ratio, which indicates the proportion of PECAM-1 in the thrombus and provides a precise description of human platelet activity (in vitro). The aim of this study was to determine whether the PECAM-1/thrombus ratio could serve as a predictive factor for bleeding events during off-pump coronary artery bypass grafting (OPCAB). To achieve this, we collected blood samples from 20 patients scheduled to undergo OPCAB surgery. We assessed the PECAM-1/thrombus ratio by evaluating thrombus formation on collagen fibers under flow conditions. Subsequently, we compared the ability of the PECAM-1/thrombus ratio in predicting bleeding risk with other methods that evaluate hemostasis activity. These methods included assessing platelet P-selectin secretion, platelet exposure of phosphatidylserine, plasma coagulation and fibrinolysis system activity, and thrombus formation using the T-TAS assay. Our findings revealed a positive correlation between the PECAM-1/thrombus ratio and the amount of blood component units transfused (BCUT) during the OPCAB surgery. Furthermore, BCUT did not show any significant correlation with other measured hemostasis parameters. This preliminary study suggests that the PECAM-1/thrombus ratio might be a good predictor of bleeding risk during the OPCAB procedure.

## 1. Introduction

Platelet endothelial cell adhesion molecule 1 (PECAM-1) is a transmembrane glycoprotein expressed on the surface of platelets, endothelial cells, monocytes, neutrophils, and lymphocytes [1]. PECAM-1 is considered an antiplatelet molecule. In an in vivo study, *Pecam1* deficient transgenic mice exhibited a prothrombotic phenotype [2]. Moreover, under in vitro conditions, the activation of PECAM-1 leads to inhibition of ADP-, cross-linked collagen related peptide-, and thrombin-mediated platelet activation pathways [3]. In our previous study, we introduced a new parameter called the PECAM-1/thrombus ratio to evaluate platelet activity. The PECAM-1/thrombus ratio shows the proportion of PECAM-1 within a thrombus formed on a surface coated with collagen under flow. Under in vitro conditions by using both mouse and human blood, we showed that an elevated amount of PECAM-1 in the thrombus correlated with increased concentrations of added acetylsalicylic acid (ASA), a commonly used antiplatelet drug, resulting in reduced platelet activity. Our new parameter proved to be more sensitive to changes in platelet activity and provided a more accurate description of the platelet activation status compared to commonly used parameters such as the thrombus area [4].

Off-pump coronary artery bypass grafting (OPCAB) is a heart coronary revascularization surgery performed on the beating heart. Unlike on-pump coronary artery bypass grafting (CABG), OPCAB is performed without the use of extracorporeal circulation. This approach is considered less invasive as it mitigates side effects associated with extracorporeal circulation, such as systemic inflammatory response syndrome, renal dysfunction, and respiratory complications [5]. The OPCAB procedure alters hemostatic balance, leading to reduced thrombin generation and attenuated fibrinolysis [6]. Disturbances in hemostasis can often result in bleeding, necessitating the transfusion of blood components. Bleeding, if left uncontrolled, can lead to irreversible end-organ damage, cardiovascular events, or even death [7]. Despite being a common complication, there is currently no approved method to assess bleeding risk during OPCAB. Therefore, finding a marker that can reliably predict intraoperative bleeding becomes particularly crucial. In case of suspected bleeding, preventive measures can be taken, such as modifying antiplatelet therapy or preparing patients differently for surgery.

Therefore, the aim of the study was to evaluate whether the PECAM-1/thrombus ratio could be utilized in the assessment of the bleeding risk during OPCAB. The ability to use the PECAM-1/thrombus ratio to evaluate bleeding risk will be compared with other methods that assess hemostasis activity. These methods include evaluating platelet P-selectin secretion, platelet exposure of phosphatidylserine (PS), the activity of the plasma coagulation system, the activity of the fibrinolysis system, and thrombus formation on tissue factor and collagen. 

## 2. Results

### 2.1. Patients Characteristics

Table 1 shows the patients’ characteristics included in the study. All patients were diagnosed with coronary artery disease and qualified for elective OPCAB surgery. According to Spearman’s correlation coefficient analysis, we observed a negative correlation between mean platelet volume (MPV) and the amount of blood component units transfused (BCUT) to patients during the OPCAB procedure (Figure 1, *r* = −0.445, * *p* < 0.05). However, no significant correlations were found between BCUT and the other parameters investigated. On the day of the OPCAB procedure, all patients were on acetylsalicylic acid (ASA). Patients did not have any complex and redo operations, severe renal insufficiency, hematological diseases, or hereditary deficiencies in platelet function. Patients were on one antiplatelet drug (ASA). 

### 2.2. Correlation between PECAM-1/Thrombus Ratio and Hemostasis Parameters

According to Spearman’s correlation coefficient analysis, we observed a significant positive correlation between the PECAM-1/thrombus ratio and the amount of BCUT to patients during OPCAB surgery (Figure 2, *r* = 0.895, *** *p* < 0.001). However, no significant correlations were found between BCUT and the other parameters tested, as shown in Table 2. 

## 3. Discussion

In our preliminary study, we have shown that the PECAM-1/thrombus ratio might be a good predictor of bleeding during the OPCAB procedure. Using a wide panel of methods to evaluate the activity of the hemostasis system, we showed that bleeding during OPCAB surgery is dependent on low platelet activity. Furthermore, among the three parameters describing platelet activity, only the PECAM-1/thrombus ratio parameter showed a correlation with BCUT.

Bleeding is caused by an imbalance between the processes that maintain blood in a fluid state and the prothrombotic processes, leading to blood loss [8]. To assess this balance, we performed a comprehensive assessment of the activity of the hemostasis system in each patient. Particular emphasis was given to specific elements involved in thrombus formation to ascertain whether the blood loss during OPCAB surgery was associated with single or multiple impairments of thrombus formation. 

The universal definition of perioperative bleeding (UDPB) in adult cardiac surgery is the scale used to determine the severity of the bleeding. This scale is based on the events occurring during surgery: delayed sternal closure, postoperative chest tube output, packed red blood cells (PRBC) transfusion, fresh frozen plasma (FFP) transfusion, platelet concentrates (PLT) transfusion, cryoprecipitate transfusion, use of coagulation factors concentrates, use of recombinant activated factor VII (rFVIIa), and surgical re-exploration. This classification comprises five classes (0–4) indicating the severity of bleeding. If the characteristic of bleeding does not meet the criteria of the one particular class, the definition of the class that assumes more severe bleeding is applied [9]. According to UDPB in adult cardiac surgery, the participants underwent mild or moderate bleeding (Class 1 and Class 2 respectively) during the OPCAB procedure. However, in our study, they did not meet all the criteria of this classification since they did not receive cryoprecipitate, rFVIIa, coagulation factors’ concentrates; they were not reoperated because of bleeding, and the chest output exceeded between 160 and 570 mL (mild bleeding: 601–800 mL). Thus, assignment to class was based on the blood components units transfused during the OPCAB procedure [9]. However, in some clinical studies, simple classification of bleeding based on RBC, PLT, FFP/Octaplas units transfused is used [10]. Therefore, in our study, we also assessed blood loss by blood components units transfused during the operation. 

According to guidelines, participants in our study were not at high risk of bleeding; therefore, monotherapy with ASA was continued [11]. Before the OPCAB procedure, two patients received rivaroxaban; one patient received clopidogrel; and one patient received ticagrelor. These drugs were withdrawn to minimize the risk of massive bleeding [11]. Therefore, there were no differences in pharmacotherapy that could significantly influence this outcome. 

Platelets play a crucial role in preventing blood loss at the site of vascular injury, forming a plug, and becoming activated. Platelet activation is a complex process comprising multiple stages [12]. In our study, we evaluated whether platelet activation under specific states affects bleeding risk. To achieve this, we utilized three methods to assess platelet activity under different activation statuses. These methods included evaluating the PECAM-1/thrombus ratio, assessment of P-selectin secretion, and monitoring PS exposure.

The formation of fibrin fibers by the plasma coagulation system strengthens the platelet plug. To avoid vessel occlusion caused by a growing thrombus, the fibrinolysis system works to dissolve the fibrin fibers, thereby maintaining the patency of the injured vessel [12]. To evaluate the activity of the plasma coagulation system, we assessed the fibrin net density in the platelet-poor plasma (PPP). Additionally, we determined the activity of the fibrinolysis system using the euglobulin clot lysis time (ECLT) assay. Finally, the overall process of thrombus formation, which is the result of the combined activity of the aforementioned elements, was performed using the Total Thrombus Formation Analysis System (T-TAS). In previous studies, it has been shown that blood loss (defined by Poston et al. as the volume of the blood collected by cell saver and chest drainage devices) during OPCAB surgery correlated with platelet activity measured through collagen-induced platelet aggregation [13]. Collagen, a main element of the subendothelial matrix, plays a crucial role in the initial phases of thrombus formation by facilitating platelet adhesion and aggregation to the exposed collagen fibers during vessel damage [12]. This, coupled with the fact that the PECAM-1/thrombus ratio is a parameter describing platelet functions, suggests that intraoperative bleeding results from decreased platelet activity, particularly impaired initial phases of thrombus formation. It is worth mentioning that the correlation coefficient between collagen-induced platelet aggregation and bleeding was weaker (*r* = 0.42) compared to the correlation coefficient between the PECAM-1/thrombus ratio and BCUT (*r* = 0.895). This indicates that the PECAM-1/thrombus ratio is more sensitive and thus an appropriate tool for the precise assessment of platelet activity. This difference in sensitivity may stem from the activation status of platelets that form the thrombus inside the flow chamber in our study. After vessel wall damage, platelets adhere and aggregate to the exposed subendothelial matrix. Subsequently, platelets undergo continuous activation triggered by elements present in the subendothelial matrix and agonists released from platelet granules. During this process, platelets change shape, and irreversible exposure of PS occurs. Platelets with exposed PS (PS-positive platelets) are considered procoagulants since PS catalyzes the activation of coagulation factors and supports fibrin formation [14]. In our previous study, we have shown that PECAM-1 is only present on the surface of platelets that do not expose PS (PS negative platelets) and thus do not undergo potent activation. To confirm that platelets were not irreversibly activated, we used DiOC6(3) staining, which penetrated only PS-negative platelets with intact cell membranes. This is evident in the confocal microscopy images where the red PECAM-1 (Figure 3) molecules and green PS-negative platelets with the intact cell membrane (Figure 3) overlay, indicating that the platelets forming the thrombus in our model are in a state consistent with their first contact with the subendothelial matrix before strong activation has occurred. Therefore, low platelet activity, as expressed by a high PECAM-1/thrombus ratio under this specific state of platelet activation, could contribute to bleeding during OPCAB surgery. The method of collagen-induced platelet aggregation is based on measuring impedance changes across two electrodes to which platelets attach and aggregate after activation with collagen [13]. However, it is possible that platelets within the aggregate are under different activation states, including both PS-negative and PS-positive platelets. Therefore, the overall value of this parameter depends on platelets that are in a state consistent with the initial contact with collagen, as well as on platelets that catalyze coagulation [15]. The presence of PS-positive platelets could lead to the formation of fibrin fibers that enhance platelet aggregation, a process dependent on the platelet integrin GPIIb/IIIa, which acts as receptors for fibrin, connecting adjacent platelets. In our study, the effect of fibrin on platelet aggregation could be omitted because an irreversible thrombin activator, PPACK, which diminishes fibrin formation, was added to the sample before blood was perfused through the flow chamber. This makes collagen-induced platelet aggregation less accurate for bleeding prediction during OPCAB surgery compared to the PECAM-1/thrombus ratio. Moreover, this hypothesis might be confirmed by the lack of correlation between the number of BCUT and the degree of P-selectin secretion and PS exposure. P-selectin is stored in the alpha granules of unstimulated platelets and is secreted to the plasma membrane upon platelet stimulation with agonists. The main role of P-selectin during thrombus formation is to bind to its receptor PSGL-1 on leukocytes, facilitating their integration into the thrombus. Furthermore, P-selectin enhances platelet aggregation and the formation of stable aggregates [16]. The platelet expression of P-selectin serves as the main marker of platelet secretion, indicating that platelets are activated enough to release substances responsible for further platelet activation processes leading to PS exposure [17]. Therefore, the secretion of P-selectin and PS exposure are not associated with the initial phase of platelet activation, whereas the PECAM-1/thrombus ratio is.

Patients with low platelet counts are at risk of intraoperative bleeding [18]. However, in our study, the platelet count fell within the normal range, ranging from 130 × 10^3^/µL to 350 × 10^3^/µL. As a result, we did not observe any correlation between PLT and blood loss during OPCAB surgery. However, we showed a negative correlation between MPV and BCUT. MPV serves as an indicator of platelet activity, and its increased value has been observed in cardiovascular diseases, cerebral stroke, and diabetes. The enhanced activity of platelets with higher MPV is attributed to the marked activation of megakaryocytes by cytokines during platelet formation and the greater content of dense granules [19].

Our study indicated that patients with lower MPV values required more BCUT during OPCAB surgery. However, it is important to note that the correlation coefficient between MPV and BCUT is weaker than the correlation coefficient between the PECAM-1/thrombus ratio and BCUT (*r* = −0.445 vs. *r* = 0.895). This finding reinforces the notion that the PECAM-1/thrombus ratio is a parameter that precisely describes platelet activity and confirms that blood loss during OPCAB surgery is due to low platelet activity. To assess whether bleeding during OPCAB surgery could be associated with an impaired overall process of thrombus formation, we used the T-TAS system in our study. For this purpose, we used the AR microchip, which is coated with collagen and tissue factor (TF). During blood perfusion through the microchip, platelets are activated by collagen, while TF activates coagulation pathways. This setup replicates physiological conditions, as the subendothelial matrix contains not only collagen but also TF, and the resulting thrombus consists of platelets and fibrin fibers [20]. Additionally, this parameter is influenced by the activity of the fibrinolytic system, as there is an immediate activation of the fibrinolytic system during thrombus formation [21]. Our findings showed that the parameters obtained from the T-TAS assay did not correlate with BCUT. This indicates that the mechanism of bleeding is more related to platelet function than to impaired overall thrombus formation. The AR chip has been previously demonstrated to be useful for bleeding prediction in patients receiving oral anticoagulants and undergoing catheter ablation. In those cases, the reason for bleeding was the mechanism of action of anticoagulants, involving the inhibition of the plasma coagulation system and fibrin formation. Furthermore, the AR chip is employed to monitor the safety of anticoagulant therapy, with the degree of attenuation of the thrombotic process on the AR chip correlating with the dose of anticoagulants. The AR chip also distinguishes different inhibitory patterns among different classes of anticoagulants (direct oral anticoagulants vs. warfarin) [22]. According to the manufacturer’s instructions, the AUC value can be an indicator of high bleeding risk. However, in our study, the AUC parameter was not a good predictor of bleeding during OPCAB surgery. This discrepancy can be attributed to the fact that patients undergoing OPCAB were on antiplatelet therapy with ASA, rendering the AUC, which is partially fibrin-dependent, not correlated with BCUT. Notably, fibrin net density, which indicates the activity of the coagulation system, and ECLT, which indicates the activity of the fibrinolysis system, also did not correlate with BCUT. These results suggest that applied pharmacology determines the direction of searching for strategies that could be implemented in bleeding prediction. However, at the current stage of our research, we are unable to determine whether the platelet activity-dependent blood loss resulted from an excessively high concentration of ASA in the blood.

There are certain limitations that are worth mentioning in our study. One of them is the relatively small number of patients who participated. Despite this limitation, the high correlation coefficient and very promising results led us to consider these findings as a starting point for assessing the utility of the PECAM-1/thrombus ratio parameter in different patient groups. While this preliminary study involved patients receiving ASA, our next study will focus on determining whether blood loss during OPCAB surgery remains platelet-dependent in patients taking anticoagulants. Additionally, we aim to assess the utility of the PECAM-1/thrombus ratio in patients undergoing on-pump CABG, where the use of extracorporeal circulation and resulting inflammatory state may lead to different changes in hemostasis compared to OPCAB patients [23,24]. These factors could potentially affect the significance of using this parameter in on-pump CABG patients. 

We showed that PECAM-1/thrombus ratio is able to predict the development of platelet-dependent or partially platelet-dependent bleeding. In case of severe bleeding associated with impairment of the multiple elements of the hemostasis system, the PECAM-1/thrombus ratio would be an insufficient marker of bleeding risk assessment because it describes only platelet activity. In case of platelet-dependent severe bleeding, the less sensitive parameters like platelet P-selectin secretion or PS exposure may be also changed. This only suggests that the PECAM-1/thrombus ratio describes and differentiates platelet activity in an extremely precise way and shows undoubted usefulness of this new indicator in platelet activity assessment. It is important to note that the PECAM-1/thrombus ratio is determined in laboratory conditions using a special assay, making routine determination, as with MPV, currently not possible. Nevertheless, it holds potential for use in preclinical studies and clinical trials evaluating the safety and effectiveness of new antiplatelet drugs. The applicability of the PECAM-1/thrombus ratio in assessing bleeding risk in different surgical procedures requires further in-depth studies. 

## 4. Materials and Methods

### 4.1. Study Population

This study was conducted as a prospective observational study at a single center, and it received approval from the Local Committee (approval number: APK.002.103.2023). The study enrolled patients aged ≥18 years, both women and men. All patients provided informed consent before being included in the study. The recruitment period lasted from March 2023 to June 2023, during which 24 patients were initially screened. However, four patients were excluded from the study due to reasons such as malignant granuloma or conversion to an on-pump CABG procedure. Blood samples for our study were drawn from the patients the day before OPCAB surgery, and sodium citrate (at a final concentration of 3.2%) was used as an anticoagulant for the blood samples. The OPCAB surgeries were performed by one team of seven experienced surgeons who specialize in this procedure. A heparin dose 1.5 mg/kg was administrated, and the activated clotting time was kept at approximately 250 s. After completion of the procedure, heparin activity was neutralized with protamine sulphate at ratio 1:1. Patients underwent different revascularization modes (LIMA-LAD, RIMA-Dg-OM1-OM2 Y-graft, Ao-RPD, SV-RCA, LRA-OM, LIMA-LAD-LAD, Ao-OM-Cx, LIMA-Dg, Ao-(RIMA)-LAD, LIMA-Cx, LRA-RPD, SV-OM2-RPD, LRA-Dg-OM, Ao-OM, Ao-OM1-Cx, AoDg-LAD, Ao-OM-RPD T graft, LIMA-Dg-LAD, RIMA-Cx Y graft, Ao-PDA), and different numbers of bypasses were performed. The chest output was measured from the moment of placement of the drains in the mediastinum until 6 AM the next day.

Medical data were collected from each patient, including age, sex, blood morphology, the number of blood components transfused during the OPCAB procedure, the antithrombotic drugs received by the patients, and data on the course of the OPCAB procedure. 

### 4.2. Blood Loss Assessment

The blood components that were transfused to the patients included packed red blood cells, leukocyte-depleted platelet concentrate, leukocyte-depleted apheresis platelet concentrate, and fresh frozen plasma. The number of transfused units (BCUT) indicated the intensity of the blood loss. Blood loss in this study was defined as the total number of BCUT in each patient during the surgery and on the day of surgery while in the postoperative Intensive Care Unit. The value of selected hemostasis parameters was correlated with the BCUT. 

### 4.3. Confocal Microscopy Observation

In the experiments involving laser-induced thrombosis and visualization of the fibrin net, we used specific equipment. The setup included a fixed-stage microscope, the Zeiss Axio Examiner Z1 (Carl Zeiss Microscopy GmbH, Oberkochen, Germany). Additionally, we utilized a confocal scanner unit (CSU-X1, Yokogawa Electric Corporation, Tokyo, Japan). The microscope was equipped with a W Plan-Apochromat 20×/1.0 water immersion objective (Carl Zeiss Microscopy GmbH, Oberkochen, Germany). For visualization purposes, we employed a confocal microscopy system to observe the thrombi formed in the flow chamber and the fibrin net in the clots. SlideBook 6.0 (Intelligent Imaging Innovations, Inc., Denver, CO, USA) was used to analyze the images from confocal microscopy.

### 4.4. Flow Chamber Characteristics and Preparation of Collagen-Coated Surfaces

The flow chamber used in the study was a transparent, polycarbonate-made, parallel-plate type with specific dimensions (height: 50 μm, width: 3 mm, length: 30 mm). It was equipped with a metal (medical steel) inlet and an outlet installed at an angle of 15° in relation to the flow axis. To create a detachable bottom for the chamber, disposable coverslips (0.2 mm thick) were employed. Before use, the coverslips were degreased in a mixture of 2 M HCl and 50% ethanol. To facilitate thrombus formation under flow conditions, the coverslips were coated with a microspot of type I collagen suspension. The collagen suspension was diluted with 1.67 mM acetic acid to achieve a concentration of 50 μg/mL, using Horm collagen from Nycomed, Zurich, Switzerland. The spot of collagen had a diameter of 1 mm. After coating, the coverslips were incubated overnight at 4 °C in a humid chamber. Subsequently, they were washed with saline to remove any unbound collagen and then blocked for 30 min with 1% (*w*/*v*) bovine serum albumin (BSA) in Hepes buffer (138 mM NaCl, 2.8 mM KCl, 8.9 mM NaHCO_3_, 0.8 mM KH_2_PO_4_, 0.8 mM MgCl_2_, 5.5 mM glucose, 3.5 mg/mL albumin, and 10 mM HEPES, pH 7.4). This blocking step aimed to minimize nonspecific interactions between blood constituents and the glass surface. Following the blocking process, the coverslips were rinsed again with saline before starting the experiment. The flow chamber was then assembled and filled with Hepes buffer containing 0.1% BSA and glucose (5.5 mM), providing the appropriate conditions for the experiments.

### 4.5. Model of Thrombus Formation in a Flow Chamber and PECAM-1/Thrombus Ratio Assessment

To inhibit coagulation during the assay, blood was treated with d-phenylalanyl-l-prolyl-l-arginine chloromethyl ketone (PPACK) (final concentration, 40 µM; Santa Cruz Biotech., Dallas, TX, USA). PPACK is a selective, irreversible thrombin inhibitor. The blood samples were then supplemented with 3,3′-dihexyloxacarbocyanine iodide (DiOC6(3)) at 1 µM, a platelet stain from Life Technologies, Molecular Probes, USA. After incubating the samples for 2 min at 37 °C to stain the platelets, they were supplemented with 10 mM CaCl_2_ and 3.37 mM MgCl_2_ (for mouse blood, both at 3 mM concentration) just before perfusion into the chamber. The whole blood was perfused into the flow chamber through silicon tubing with an inner diameter of 1 mm. The flow rate was adjusted to achieve a wall shear rate of 1000/s based on the Poiseuille equation, which reflected the conditions of arterial circulation. To allow for the full formation of the thrombus on the collagen-coated surface, the blood flow through the chamber continued for 4 min starting from the entry into the chamber. The thrombus-covered surface was then washed with Hepes buffer after this period. Following the disassembly of the flow chamber, PECAM-1 within the thrombi was stained by topically applying 2 µL of the Alexa Fluor 647-labeled PECAM-1 antibody (Alexa Fluor 647 antihuman CD31 antibody) obtained from Bio Legend, San Diego, CA, USA, onto a thrombi-rich area. The samples were incubated with the antiCD31 antibody at room temperature for 5 min and then washed to ensure that we observed PECAM-1 present on the cell surface. Furthermore, according to the manufacturer’s information, the antiCD31 antibody was specifically used for immunohistochemical staining of PECAM-1 present on the cell surface, which excludes the possibility of staining the soluble form of PECAM-1.

For end-stage measurements (including the surface area covered by the thrombus, referred to as the thrombus area) in the formed thrombi, two-color fluorescent pictures were taken using a confocal microscope (Figure 3, on the right). The subsequent analysis was carried out using SlideBook 6.0 software (Intelligent Imaging Innovation, Inc., Denver, CO, USA). The area of fluorescence from platelet PECAM-1 (Figure 3, on the left, red color) was divided by the thrombus area (Figure 3, in the middle, green color), and the resulting value was referred to as the PECAM-1/thrombus ratio.

### 4.6. P-Selectin Secretion and PS Exposure Assessment with a Flow Cytometer

In the study, the CytoFLEX Model A00-1-1102 (Beckman Coulter Inc., Brea, CA, USA) along with the CytExpert 2.4 software from the same company, was used. To prepare the blood samples, blood was centrifuged at 129× *g* for 10 min at room temperature. The platelet count in the supernatant was estimated to be 10  ×  10^7^/mL. To stain the platelets, antihuman GPIb (Alexa Fluor 488 antihuman CD42b, BioLegend, San Diego, CA, USA) was added to the platelet suspension at a final concentration of 5 µg/mL. To assess PS exposure, which is a marker of irreversible platelet activation, the sample was mixed with Alexa Fluor 647-labeled annexin V (Thermo Fisher Scientific, Waltham, MA, USA) at a final dilution of 1:15, which directly binds to PS. To assess P-selectin exposure, which is a marker of platelet secretion, the sample was mixed with an antihuman P-selectin antibody (Alexa Fluor 647 antihuman CD62P, BioLegend, San Diego, CA, USA) at a final concentration of 5 µg/mL. After the necessary preparations, the platelets were supplemented with CaCl_2_ at a final concentration of 1 mM, activated with ADP at a final concentration of 20 µM, and gated based on the binding of antiGPIb antibodies (Figure 4).

Before activation, the platelets formed a single population. However, after activation with ADP, a new population appeared with surface markers indicating platelet activation. The degree of P-selectin secretion or PS exposure was assessed as a percentage of P-selectin-positive or PS-positive platelets in the total platelet population.

### 4.7. Assessment of Fibrin Net Density in Platelet-Poor Plasma (PPP)

To evaluate the fibrin net density in clots, a method described previously [25] was employed. First, blood samples were centrifuged to obtain platelet-rich plasma (PRP) by centrifuging at 200× *g* for 20 min. Subsequently, PPP was obtained by further centrifuging the PRP at 14,000× *g* for 5 min. Alexa Fluor 488-labeled human fibrinogen (Fibrinogen from Human Plasma, Alexa Fluor™ 488 Conjugate, Thermo Fisher Scientific, Waltham, MA, USA USA) was added to the PPP at a final concentration of 15 µM. To induce clot formation, CaCl_2_ was added at a final concentration of 20 mM. The samples were then incubated at 37 °C for 2 h to allow clot formation. To assess relative clot density, images of the resultant clots were obtained (Figure 5). In each image, four 50 µm long straight lines were placed randomly. The number of fibrin fibers crossing each line was counted. The average of the resultant values was then referred to as the relative clot density.

### 4.8. Assessment of Euglobulin Clot Lysis Time

ECLT was used to determine the time of clot dissolution. It was measured following the method described by Tomczyk et al. [26] with some modifications [25]. To obtain the euglobulin fraction, 200 µL of PPP were mixed with 3800 µL of 0.016% acetic acid, adjusted to pH 4.65. The resulting mixture was then incubated for 1 h at 4 °C and subsequently centrifuged at 4000 rpm for 10 min at 4 °C. This process yielded the euglobulin fraction, which was then dissolved in 200 µL of Tris buffer with a pH of 7.4. For the ECLT assay, 150 µL of the euglobulin fraction were placed in a well of a microplate. Clot formation was initiated by adding 50 µL of CaCl_2_ solution in Tris buffer with a pH of 7.4 to the euglobulin fraction, resulting in a final CaCl_2_ concentration of 10 mM in the well. The changes in optical density in the wells were recorded at a wavelength of 405 nm at 37 °C, using the microplate factor ELx808, at 1-min intervals. The ECLT was defined as the time elapsed from the addition of the CaCl_2_ solution to the well (initial absorbance value) to the point where the minimal absorbance value was achieved due to clot lysis.

### 4.9. Assessment of the Thrombus Formation Process with Total Thrombus Formation Analysis System

In our study, we utilized the T-TAS^®^01 Instrument (Zacros, Tokyo, Japan). T-TAS is a microchip-based assay specifically designed to analyze the process of thrombus formation under flow conditions. The assay employs an AR microchip coated with type I collagen and tissue factor on the blood flow path. For the experiments, citrated blood (480 µL) was mixed with a CaCl_2_ solution (12 mM) containing corn trypsin inhibitor (CTI, 50 μg/mL; Zacros, Tokyo, Japan), which serves as an inhibitor for the XII factor of coagulation. The blood was then perfused throughout the capillary over the microchip, with the shear stress estimated to be 600/s. The thrombus formation in the microchip is a result of both primary (platelet-dependent) and secondary (plasma coagulation system-dependent) hemostasis. As the thrombus grows, it leads to an increase in the pressure inside the flow path capillary. The pressure changes were registered by the pressure sensor located upstream in the chamber. The resulting graph presented the thrombus formation by depicting the pressure changes over time due to the formation of the thrombus. From the graph, three important parameters can be obtained:○Occlusion Start Time: This refers to the lag time for the flow pressure to reach 10 kPa due to the partial occlusion of the capillary.○Occlusion Time: This indicates the lag time for the flow pressure to reach 60 kPa from the baseline pressure.○Area under the Curve: This represents the area under the flow pressure vs. time curve and provides information about the overall thrombus formation.

### 4.10. Statistical Analysis

Spearman’s correlation analysis was performed using GraphPad Prism 9.4.1 software (GraphPad Software, San Diego, CA, USA). We correlated the obtained values of hemostasis parameters, age, and blood morphology with the total number of BCUT in patients during OPCAB surgery. Statistical significance was determined using a *p*-value threshold of <0.05, and the significance levels were denoted as follows: * for *p* < 0.05, ** for *p* < 0.01, and *** for *p* < 0.001.

## Figures and Tables

**Figure 1 ijms-24-13254-f001:**
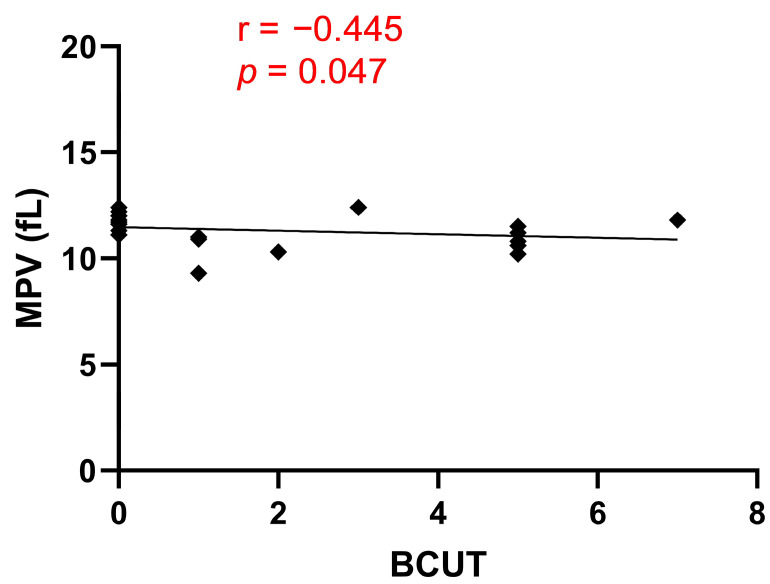
Correlation between MPV and BCUT; *n* = 20.

**Figure 2 ijms-24-13254-f002:**
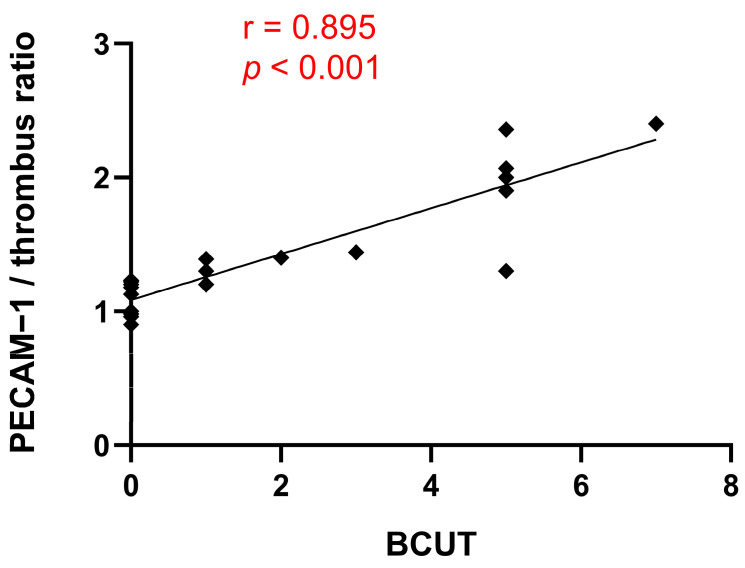
Correlation between PECAM−1/thrombus ratio and BCUT; *n* = 20.

**Figure 3 ijms-24-13254-f003:**
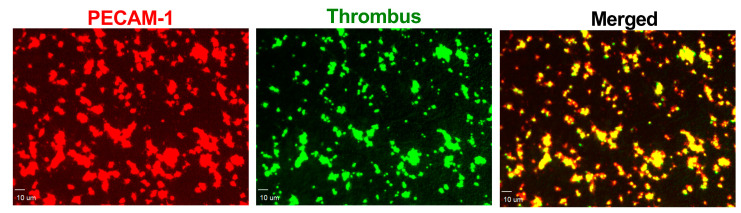
Representative pictures from confocal microscopy of thrombus formed on the collagen fibers in the flow chamber. **Left** (red): platelet PECAM-1 visualized with human anti-PECAM-1 antibody. **Middle** (green): thrombus composed of platelets visualized with DiOC6(3). **Right** (merged): the overlapped pictures of platelet PECAM-1 and thrombus. Bar = 10 µm.

**Figure 4 ijms-24-13254-f004:**
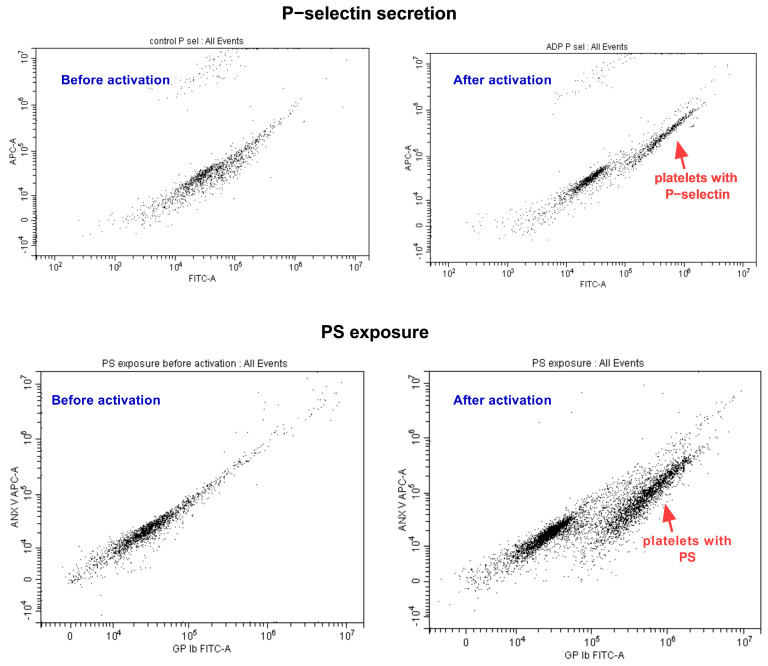
Representative pictures from flow cytometry. The upper panel shows platelets that undergo P−selectin secretion. Before platelet activation (on the **left**), only one platelet population is seen. After activation (on the **right**), platelet population with secreted P−selectin appears (indicated with red arrow). The bottom panel shows platelets that undergo PS exposure. Before platelet activation (on the **left**), only one platelet population is seen. After activation (on the **right**), platelet population with exposed PS appears (indicated with red arrow).

**Figure 5 ijms-24-13254-f005:**
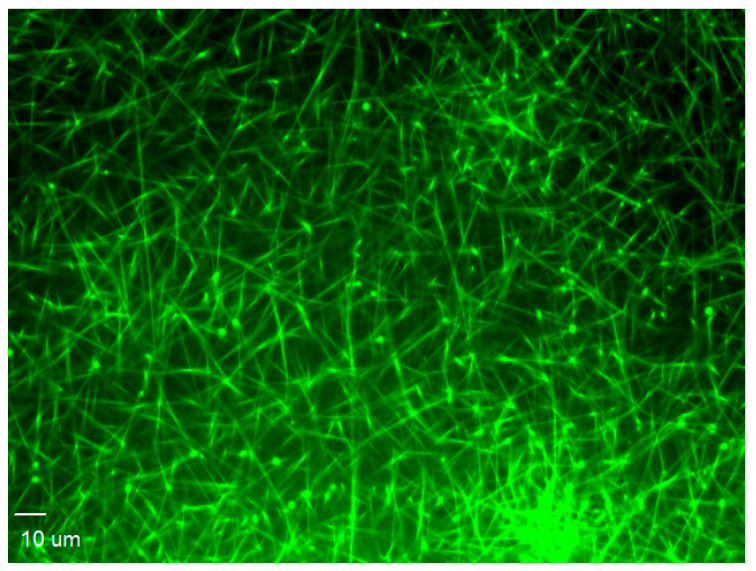
Representative picture from confocal microscopy of fibrin net in clot formed after recalcification of PPP. Bar = 10 µm.

**Table 1 ijms-24-13254-t001:** Patients’ characteristics. Data are presented as medians (interquartile ranges) and as the absolute values (number of bypasses, number of BCUT). The table shows correlations between age, blood morphology parameters, number of bypasses and BCUT (included: packed red blood cells, leukocyte-depleted platelet concentrate, leukocyte-depleted apheresis platelet concentrate, and fresh frozen plasma); * *p* < 0.05.

Characteristics	Patients (*n* = 20)	*r*	*p*
Sex	Male (85%)	-	-
Aspirin	100%	-	
Age (years)	65 (59.71)	−0.27	0.26
Number of BCUT			
0	9 (45%)	-	-
1	3 (15%)	-	-
2	1 (5%)	-	-
3	1 (5%)	-	-
5	5 (25%)	-	-
7	1 (5%)	-	-
Number of bypasses	2–5	−0.06	0.81
2 bypasses	4 (20%)	-	-
3 bypasses	10 (50%)	-	-
4 bypasses	4 (20%)	-	-
5 bypasses	2 (10%)	-	-
Chest output (mL)	380 (268, 510)	−0.05	0.83
White blood cells (WBC, ×10^3^/µL)	6.86 (5.89, 8.44)	−0.06	0.80
Red blood cells (RBC, ×10^6^/µL)	4.58 (4.23, 4.81)	−0.21	0.38
Hemoglobin (HGB, g/dL)	14.35 (13.15, 14.98)	−0.28	0.24
Hematocrit (HCT, %)	41.2 (39.53, 43.74)	−0.23	0.33
Platelet count (PLT, ×10^3^/µL)	214.8 (193.5, 334.0)	0.05	0.85
Mean platelet volume (MPV, fL)	11.40 (10.83, 11.95)	−0.45	* 0.047

**Table 2 ijms-24-13254-t002:** Correlations between blood components units transfused (BCUT) during OPCAB surgery and hemostasis parameters; *n* = 20. T-TAS—Total Thrombus Formation Analysis System, AUC—Area Under the Curve, OST—Occlusion Start Time, OT—Occlusion Time, ECLT—Euglobulin Clot Lysis Time, PS—phosphatidylserine.

Parameter	*r*	*p*
T-TAS parameters:		
AUC	−0.07	0.78
OST	0.06	0.8
OT	0.07	0.79
Fibrin net density	−0.37	0.13
ECLT	−0.06	0.82
P-selectin expression	0.38	0.18
PS-exposure	0.25	0.38

## Data Availability

The data that support the findings of this study are available on request from the corresponding author. The data are not publicly available due to privacy or ethical restrictions.

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
