# Peer review of "PECAM-1/Thrombus Ratio Correlates with Blood Loss during Off-Pump Coronary Artery Bypass Grafting (OPCAB) Surgery: A Preliminary Study"

_ijms, 2023, doi:10.3390/ijms241713254_

Round 1

Reviewer 1 Report

The aim of this study was to determine whether the PECAM-1/thrombus ratio could serve as a predictive factor for bleeding events during off-pump coronary artery bypass grafting (OPCAB).

In my opinion, OPCAB strategy (ACT levels, heparin and protamine dose, numbers of bypass performed) must be stated, together with a more punctual evalutation of BCUT infused, since administration of different blood products may have different causes and therefore may be a bias in this analysis. Preoperative data such as hemoglobin, platelets count and coagulative assessment should be elicited, since the population studied is small. Number of BCUT for patients must be specified, since it appears that a conspicuous number of patients required a discrete number of blood products: it should be evident if it may be linked only to the PECAM-1/thrumbus ratio or if there are some preoperative and/or intraoperative factors that may influence this outcome. 

In conclusion, since this study is focused on a peculiar population, in my opinion, the characteristics of this group must be addressed adequately in order to highlight the link between PECAM-1/thrumbus ratio and bleeding risk.

Quality of English language: good.

Reviewer 2 Report

I reviewed with interest the manuscript of Bielicka et al. "PECAM-1/thrombus ratio correlates with blood loss during off-pump coronary artery bypass grafting (OPCAB) surgery: A preliminary study". In this article, the authors tried to evaluate the possibilities of using their new indicator - PECAM-1/thrombus ratio - in predicting possible blood loss during off-pump coronary artery bypass grafting surgery. According to the authors, it was the new indicator that most correlated with blood loss during OPCAB surgery. This is a new scientific fact that deserves attention and further research. Nevertheless, when reviewing, I had questions and comments that I would like to receive answers from the authors.

1. As an estimate of the volume of blood loss, the authors used the amount of blood component units transfused, which is a very inaccurate method for assessing blood loss. First, the volume of blood loss does not necessarily correlate with the volume of infusion. Secondly, the management of blood loss can vary greatly between institutions, depending on the treatment protocol used (1), and depending on the preferences of the ICU physician. Perhaps more clinically relevant would be a bleeding score according to one of the bleeding classifications (2). If the PECAM-1/thrombus ratio were to be able to predict the development of serious perioperative bleeding, then this would show the undoubted usefulness of this new indicator.

2. Because the volume of blood loss during OPCAB surgery depends not only on baseline hemostasis, but also on many clinical factors (duration of surgery, number of bypasses, perioperative therapy received, baseline hemoglobin and erythrocyte levels, etc.) (3,4) , then in such a study it is necessary to take into account these indicators.

References:

1.      Silva PG, Ikeoka DT, Fernandes VA, Lasta NS, Silva DP, Okada MY, Izidoro BA, Garcia JC, Baruzzi AC, Furlan V. Implementation of an institutional protocol for rational use of blood products and its impact on postoperative of coronary artery bypass graft surgery. Einstein (Sao Paulo). 2013 Jul-Sep;11(3):310-6. doi: 10.1590/s1679-45082013000300009.

2.      Xi Z, Gao Y, Yan Z, Zhou YJ, Liu W. The Prognostic Significance of Different Bleeding Classifications in off-pump coronary artery bypass grafting. BMC Cardiovasc Disord. 2020 Jan 10;20(1):3. doi: 10.1186/s12872-019-01315-0.

3.      Kim HH, Lee KJ, Kang DR, Lee JH, Youn YN. Hemostatic efficacy of a flowable collagen-thrombin matrix during coronary artery bypass grafting: a double-blind randomized controlled trial. J Cardiothorac Surg. 2023 Jun 15;18(1):193. doi: 10.1186/s13019-023-02196-3.

4.      Wang LH, Wang XH, Tan JC, He LX, Fu RQ, Lin Y, Yao YT. Levosimendan administration is not associated with increased risk of bleeding and blood transfusion requirement in patients undergoing off-pump coronary artery bypass grafting: a retrospective study from single center. Perfusion. 2023 Mar;38(2):270-276. doi: 10.1177/02676591211049022.

No comments

Round 2

Reviewer 2 Report

The authors answered my questions in detail and made appropriate changes to the text of the manuscript. I have no other comments.

No comments